# Association between Cerebral Coordination Functions and Clinical Outcomes of Alzheimer’s Dementia

**DOI:** 10.3390/brainsci12101370

**Published:** 2022-10-09

**Authors:** Yuan-Han Yang, Ying-Han Lee, Chen-Wen Yen, Ling-Chun Huang, Yang-Pei Chang, Ching-Fang Chien

**Affiliations:** 1Department of Neurology, Kaohsiung Medical University Hospital, Kaohsiung Medical University, Kaohsiung 80708, Taiwan; 2Neuroscience Research Center, Kaohsiung Medical University, Kaohsiung 80708, Taiwan; 3Department of Neurology, Kaohsiung Municipal Ta-Tung Hospital, Kaohsiung Medical University Hospital, Kaohsiung 80145, Taiwan; 4Department of and Master’s Program in Neurology, Faculty of Medicine, Kaohsiung Medical University, Kaohsiung 80708, Taiwan; 5Post-Baccalaureate Medicine, Kaohsiung Medical University, Kaohsiung 80708, Taiwan; 6Department of Mechanical and Electro-Mechanical Engineering, National Sun Yat-sen University, Kaohsiung 80424, Taiwan

**Keywords:** Alzheimer’s dementia, kinetic, depth sensor, clinical dementia rating, cognitive ability screening instrument

## Abstract

Background: Alzheimer’s dementia (AD) is a degenerative disease that impairs cognitive function, initially, and then motor or other function, eventually. Motor coordination function impairment usually accompanies cognition impairment but it is seldom examined whether it can reflect the clinical outcomes of AD. Methods: 113 clinically diagnosed AD patients with a mean age of 78.9 ± 6.9 years underwent an annual neuropsychological assessment using the Mini-Mental State Examination (MMSE), the Cognitive Abilities Screening Instrument (CASI), the Sum of Boxes of Clinical Dementia Rating (CDR-SB), and the CDR. The cerebral coordination function was evaluated through correlations among 15 joints with a kinetic depth sensor annually. An intra-individual comparison of both cognitive and motor coordination functions was performed to examine their correlations. Results: The changes in coordination function in the lower limbs can significantly reflect the clinical outcomes, MMSE (*p* < 0.001), CASI (*p* = 0.006), CDR (*p* < 0.001), and CDR-SB (*p* < 0.001), but the changes in upper limbs can only reflect the clinical outcome in CDR (*p* < 0.001). Conclusions: The use of a kinetic depth sensor to determine the coordination between joints, especially in lower limbs, can significantly reflect the global functional and cognitive outcomes in AD. Such evaluations could be another biomarker used to evaluate non-cognitive outcomes in AD for clinical and research purposes.

## 1. Introduction

Alzheimer’s dementia (AD) is recognized as a chronic, progressively neurodegenerative disease which does not currently have a cure. One of the main pathological findings is lots of cerebral amyloid-β [1,2] which leads to changes such as aggregation of tau protein, decreased cerebral metabolism, and eventual neuron loss [3]. Such pathological changes lead to a decline in cognitive abilities, the occurrence of behavioral and psychiatric syndromes, and the global functional loss, which can require institutionalization at advanced stages [4]. Recent studies have indicated that the deposition of cerebral amyloid might start from the default mode network (DMN) and that the frontoparietal, frontotemporal, and other cerebral areas then have subsequent clinical manifestations corresponding to these pathologically damaged areas [5]. Along with the course of AD, amyloid load or tau protein aggregation may plateau when AD is in its clinical stage [5,6]. In other words, impaired cerebral function in clinical AD may not be limited to cognitive function.

Cerebral function consists of highly complex integrations between the motor, sensory, and coordination systems to manage interactions between interior and exterior stimuli [7,8]. Impairment of these circuits leads to various clinical presentations; however, some of them might be minor and are not easily recognized in the early stages of AD. To date, the diagnostic criteria for AD have mostly used impaired cognitive function as the core criteria, together with other supporting information or biomarkers to increase the diagnostic accuracy. It does not typically indicate that other non-cognitive functions could still be preserved when and after the diagnosis is made.

The continuously pathological process of developing AD started from amyloid aggregation and deposition, tau protein formation, and cellular death in the cerebral cortex responsible for cognitive function and eventually extended to those of non-cognitive function. There have been increasing reports of evaluating non-cognitive symptoms, such as a loss of motor function in AD [9,10]. Impaired coordination function and motor symptoms and signs have been observed in 15–50% of AD patients [11,12,13]; they can also be used to predict cognitive and functional decline [11] and correlate with the deposition of cerebral amyloid-β [14].

These motor non-cognitive impairments indicate the importance of amyloid-mediated degeneration of the cholinergic system in AD [10]; however, few quantitative studies have previously examined this process [15,16]. Recent advances in neurophysiologic testing are being used to elucidate the complex processes necessary to ensure accurate movements. Integration of a wide range of sensory and visuospatial information is essential for accurate movement analysis, such as postural control derived from several brain regions [7,8,17]. However, more objective and precise measuring and analysis are needed to reflect and report on these non-cognitive functions. Our previous study used kinetic depth sensors to assess postural stability and joint coordination in a young and aged population; we found that aging increases the coupling strength, decreases the changing velocity, and reduces the complexity of the inter-joint coordination patterns [18]. AD is a continuously degenerative course starting in cognitive function to other varied cerebral functions beyond cognition. These motor non-cognitive functions are easily ignored in routine practice, especially for the minor change, and they are potentials to be considered as part of AD clinical course. To provide a more objective and precise evaluation of the changes in coordination function from these joints to reflect the AD clinical course, we are going to examine the association between clinical outcomes by psychometrics and changes in joint correlations by a correlation coefficient of joints from kinetic depth sensor through the intra-individual change’s manners.

## 2. Materials and Methods

### 2.1. Patients

All patients diagnosed with AD at the Department of Neurology, Kaohsiung Municipal Ta-Tung Hospital were recruited in a longitudinal cohort project to trace the clinical outcomes of AD. A diagnosis of AD was based on the NINCDS-ADRDA criteria [19] with reference to a series of comprehensive neuropsychological tests, including the Taiwan version Mini-Mental State Examination (MMSE) derived from the Cognitive Abilities Screening Instrument (CASI) [20,21,22], the Neuropsychiatric Inventory (NPI) [23], and the Clinical Dementia Rating (CDR) scale [24]. Neuroimaging and blood checks were performed simultaneously to exclude other conditions which could contribute to a diagnosis of AD.

### 2.2. Evaluations

All procedures were approved by the Kaohsiung Medical University Hospital Institutional Review Board (KMUHIRB-SV(I)-20190025), and written informed consent was obtained from all participants or their legal representatives prior to their inclusion within the study. For each recruited AD patient, a series of neuropsychological assessments, including the MMSE, NPI, CASI, and CDR were administered every 12 months to trace their clinical outcomes. The MMSE, NPI, CASI, and CDR were conducted by a senior neuropsychologist and an experienced physician based on information from a knowledgeable collateral source (usually a spouse or adult child). An intra-individual comparison of psychometrics to represent the cognitive outcome was used in the study. Clinical outcomes after one year were defined as either ‘improved’ or ‘worse’, and these definitions were made according to the changes between the two measured parameters for CASI, MMSE, CDR-SB, and CDR, independently. A patient where the 2nd CASI-1st CASI was ≥0, the 2nd MMSE-1st MMSE was ≥0, the 2nd CDR-SB-1st CDR-SB was ≤0, or the 2nd CDR-1st CDR was ≤0 was defined as having an improved status, and vice versa.

### 2.3. Apolipoprotein E (APOE) Genotyping

For every AD patient, restriction enzyme isotyping of the APOE allele was performed following a modification of the protocol developed by Pyrosequencing; a detailed method has been published in our previous study [25]. Individuals with one or two copies of the APOE4 allele were considered to be APOE4 positive (APOE4(+)), with all others considered APOE4 negative (APOE4(−)).

### 2.4. Posture and Joint Angle Assessments for Coordination Function

Together with the neuropsychological assessment, the Kinect Depth Sensor System was used to measure the angle formed from all 15 assessed joints, and the limb to the central axis on an annual basis. Coordination function was also evaluated via the correlation between the 2 angles through correlation coefficient variables analysis for the upper and lower limbs, separately. An intra-individual comparison of change between the two measured coefficient variables with an interval of one year was used to represent the change in cerebral coordination function. Improved coordination function was defined by a decrease in the correlation coefficient compared to its previous value, with a worse coordination function defined as the opposite.

### 2.5. Kinect Depth Sensor System

A Microsoft Kinect sensor (Microsoft Corp., Redmond, WA, USA) was connected to a personal computer-based signal processing system. The Microsoft Kinect V2 sensor, also known as the Xbox One Kinect, was used to capture joint information for a participant (25 joint centers @30 Hz). In this study, the Microsoft Kinect Software Development Kit (SDK) 2.0 (Microsoft Corp., Redmond, WA, USA)was used to obtain the spatial location of 25 human joints in three dimensions [18]. However, due to the fact that participants were standing still and not walking for gait analysis, the recruited data from some joints were not entered into the data analysis; this procedure has been validated in our previous study. In total, signals from 15 joints were included in the statistical analysis [18].

### 2.6. Standing Still Position for Data Receiving and Processing

Three 40 s test sessions were performed for each subject. In each session, the participants were instructed to look straight at a visual reference and stand still (with their arms by their sides) in a comfortable stance for 40 s. The distance between the visual reference and the test subject was about 2 m. As shown in Figure 1, the 15 joints included in this study were the (1) head, (2) neck, (3) shoulder center, (4) left shoulder, (5) right shoulder, (6) trunk center, (7) left elbow, (8) right elbow, (9) hip center, (10) left hip, (11) right hip, (12) left hand, (13) right hand, (14) left knee, and (15) right knee. 

### 2.7. Angles Formed by the Limbs to the Central Axis

The angles formed by these 15 joints lead to 20 angles from the upper limbs (10 on each side) and 10 from the lower limbs (5 on each side), making 30 angles in total.

The right upper angles were angle (∠) 4-1-7, ∠4-1-12, ∠4-2-7, ∠4-2-12, ∠4-3-7, ∠4-3-12, ∠4-6-7, ∠4-6-12, ∠4-9-7, and ∠4-9-12. The left upper angles were ∠5-1-8, ∠5-1-13, ∠5-2-8, ∠5-2-13, ∠5-3-8, ∠5-3-13, ∠5-6-8, ∠5-6-13, ∠5-9-8, and ∠5-9-13. In the upper limbs, an angle will have 19 correlation coefficients, and the second angle will lead to a correlation coefficient between them, so there will be 380 correlation coefficients for all 20 angles. Similarly, the right lower angles were ∠10-1-14, ∠10-2-14, ∠10-3-14, ∠10-6-14, and ∠10-6-14, and there will be 90 correlation coefficients in total for these 10 angles. Given to the correlation coefficients recruited from both sides are symmetric, only 190 correlation coefficients from upper limbs and 45 correlation coefficients from lower limbs will enter the statistical analysis.

Briefly, the In the Kinetic sensor will record 1200 coordination for each joint of the patient (30 Hz with 40 s). We picked the mean value of the total 1200 coordination as the representative angle of each AD patient. With a total of 113 AD patients, each patient will have 20 representative joint angles and each angle will do Pearson correlation coefficient analysis with the other 19 angles. We then evaluated the change in Pearson correlation coefficient between 1st and 2nd examinations of a total of 113 AD patients after one year to see if the change improved or worse to match the outcomes measured by MMSE, CASI, CDR, and CDR-SB. For each Pearson correlation coefficient analysis, the equation was as follows:$$r=\frac{\sum \left(x_{i}-\bar{x}\right)\left(y_{i}-\bar{y}\right)}{\sqrt{\sum {\left(x_{i}-\bar{x}\right)}^{2}\sum {\left(y_{i}-\bar{y}\right)}^{2}}}\;,for\;i=1\;to\;113$$
where r=Pearson correlation coefficient

$x_{i}=x\;\mathrm{variable}\;\mathrm{samples}$, $\bar{x}=\mathrm{mean}\;\mathrm{of}\;\mathrm{values}\;\mathrm{in}\;x\;\mathrm{variable}$

$y_{i}=y\;\mathrm{variable}\;\mathrm{samples}$, $\bar{y}=\mathrm{mean}\;\mathrm{of}\;\mathrm{values}\;\mathrm{in}\;y\;\mathrm{variable}$.

### 2.8. Statistical Analysis

Analyses were performed using the SPSS (Standard version 11.5.0; SPSS Inc., Chicago, IL, USA). All statistical tests were two-tailed, and *p* > 0.05 was considered to indicate a statistically significant difference. Age, education, CASI, MMSE, and CDR-SB, were treated as continuous variables, while sex, APOE4 status, and CDR were treated as categorical variables. The chi-square examination was used to compare the clinical outcome by CDR groups. Paired *t*-tests for the two independent groups for baseline demographic characteristics and psychometrics were used to assess differences. The correlation coefficient between two angles in the upper and lower limbs was calculated by Pearson’s correlation coefficient analysis. Similarly, correlation coefficients between the mean ratio of each angle with improved and worse coordination corresponding to their clinical outcomes were also calculated by Pearson’s correlation coefficient analysis.

## 3. Results

A total of 113 AD patients with a mean age of 78.9 ± 6.9 years were recruited into the statistical analysis and they were female predominant (66.4%). The differences between each psychometric, including MMSE, CASI, CDR-SB, and CDR, were all significantly different after a one-year follow-up (*p* < 0.001). More detailed information is illustrated in Table 1. The clinically therapeutic outcomes were measured according to several different parameters as stated above. 

In general, the ratio of improved patients ranged from 38.9% to 83.2% according to the different parameters, and the improved group was female predominant (shown in Table 2).

After the two kinetic assessments were conducted to measure the joint angle, each correlation coefficient was calculated between the two measurements for all 20 angles from the upper limbs to give a total of 380 correlation coefficients. Similarly, 90 correlation coefficients were obtained from the 10 angles formed by the lower limbs. The coordination function was assessed according to the changes between the two correlation coefficients in the upper (shown in Table 3) and lower limbs (shown in Table 4). The association between the changes in coordination function and clinical outcomes, according to different parameters, were also determined in the upper (shown in Table 3) and lower limbs (shown in Table 4). In the upper limbs, each angle will have 19 correlation coefficients for coordination function, compared with nine in the lower limbs. The ratio of improved or decreased coordination function for an angle was obtained by the number of improved or decreased coordination coefficients divided by 19 for the upper limbs and nine for the lower limbs.

In all 20 angles from the upper limbs, when the clinical outcome was defined as a change in CASI score, the mean ratio of improved coordination function was 0.784 ± 0.124 while the mean ratio for the worse coordination function group was 0.495 ± 0.137. The correlation of the mean ratio between the two groups was not significant (*p* = 0.131) (shown in Table 3). When the parameter of clinical outcome was MMSE, the mean ratio of angles with improved coordination was 0.616 ± 0.133 compared with 0.458 ± 0.154 in the worse MMSE group. The correlation between the two groups was also not significant (*p* = 0.595). Using CDR-SB as the parameter of clinical outcomes, the mean ratio of angles with improved coordination was 0.726 ± 0.126 compared with 0.668 ± 0.121 in the worse CDR-SB function group. The correlation of the mean ratio between the two groups was not significant (*p* = 0.306). Interestingly, the mean ratio of angles with improved coordination was 0.516 ± 0.188 in the improved CDR group, compared with 0.027 ± 0.010 in the worse CDR group. The correlation between these two groups was significant (*p* < 0.001). It meant the changes in coordination in the upper limbs measured via our kinetic assessment and computation, were significantly associated with the clinical outcomes evaluated by CDR.

In the lower limbs, when the clinical outcome was defined as a change in CASI score, the mean ratio of angles with improved coordination was 0.844 ± 0.182 compared with 0.533 ± 0.233 in the worse coordination CASI group. The correlation of the mean ratio between the two groups was significant (*p* = 0.006). When the parameter of clinical outcomes was MMSE, the mean ratio of angles with improved coordination was 0.333 ± 0.174 compared with 0.037 ± 0.019 in the worse MMSE group. The correlation between the two groups was also significant (*p* < 0.001). Using CDR-SB as the parameter of clinical outcomes, the mean ratio of angles with improved coordination was 0.266 ± 0.150 compared with 0.030 ± 0.017 in the worse CDR-SB group, which were also significantly correlated (*p* < 0.001). Moreover, the mean ratio of angles with improved coordination was 0.511 ± 0.297 in the improved CDR group, and 0.057 ± 0.033 in the worse CDR group. The correlation between these two groups was also significant (*p* < 0.001). The findings of the lower limbs indicate that changes in coordination function for the lower limbs, as measured by kinetic assessment and analysis, can significantly reflect the clinical outcomes as defined by MMSE, CASI, CDR, and CDR-SB (shown in Table 4).

## 4. Discussion

We have successfully identified the associations between joint coordination function and clinical outcomes in AD patients using our kinetic depth sensor assessment and analysis. The change in coordination function in the lower limbs can directly and significantly reflect the clinical outcomes, cognitive function and global function, as defined by MMSE, CASI, CDR, and CDR-SB. The change in coordination function for the upper limbs can significantly reflect the global function, as defined by CDR. Together with cognitive evaluation for AD, coordination function evaluated by kinetic depth sensor could objectively reflect the clinical outcomes of AD. We used the intra-individual change’s manner to reflect the clinical outcome of AD, which is more practicable to other case-control studies because the diagnosis of AD from the standard criteria [19] indicated the change of cognitive function should be compared to one’s previous level, not to other individual’s level.

Several previous studies have tried to report on the motor or coordination function problems which can accompany AD, but with limited results due to their lack of a sufficient and objective measurement for these conditions, especially for artificial intelligence computing. A previous study reported that motor performance contributes to functional impairments in AD, independent of cognitive impairment [26]. However, in that study, the measurements for motor function mainly depended on motor performance using several basic tests: a 360-degree turn, a measured walk, and repeated chair stands [27,28] for lower limbs and finger tapping and Purdue pegboard tests [29,30] for upper limbs. These simple clinical observations may not be sufficient to precisely and objectively assess the motor and coordination function. Moreover, these tests were usually performed only once to determine the motor or coordination function, which has limitations when evaluating these complex activities. In addition, in that study the functional impairment was measured using three standard self-report scales [31]; however, the self-report scale relies on an adequate and accurate informant. The findings of a self-evaluation may differ from the actual characteristics of the informant and may not be completed objectively.

These weaknesses were compensated for in our study by using artificial intelligence to objectively measure the movement of each joint. In our kinetic analysis for coordination function, we used 30 Hz signal detection for 40 s so we had 1200 measurements for each joint and could detect minor motions for each joint, which are usually overlooked by the naked eye without such devices [18]. Our evaluations (1200 measurements for each joint) can provide more information and data relating to motor and coordination functions. Our analysis not only examined the change in each joint, we also examined the interaction of each joint through the angles they formed and compared this over a year time period. In other words, we indeed evaluated the coordination function (except for motor function).

AD-accompanying motor dysfunction, such as bradykinesia, has been previously reported and is thought to be associated with cholinergic deficits from its pathogenesis [32]. However, like other studies, it did not precisely report how the impairments in motor function were measured. The methods we used have also been validated in a previous study [33] which reported some interesting findings.

That study used kinetic sensors to examine gait problems and concluded that impaired gait function was associated with AD [33]. The authors used the same device as us to receive the signals from joints for the gait analysis and reported that AD patients had slower gait speed, shorter step lengths, and lower step frequency. However, these reports were calculated simply by using signals from the kinetic sensor at a given spatial position, time and frequency for each joint, and lacked a more comprehensive assessment like our analyses.

Our study did have some limitations which should be addressed in future studies. Firstly, the interval between clinical outcome assessments and the kinetic analyses was only one year. If we extended the clinical course to more than 1 year, we do not know whether these results would remain the same. In other words, although so far we have promising findings that kinetic assessment for coordination function, especially in the lower limbs, could reflect clinical outcomes, it is unclear whether these findings will be maintained in the long term. More long-term analysis should be conducted in a continuous cohort study. We did not examine the possible effects of APOE genetic status or other fluid biomarkers on clinical outcomes in these patients although, to our knowledge, there was no definite association reported for these issues. These issues could be examined in the future in a study with a large sample size and a more comprehensive study design.

The clinical outcomes for AD varied according to the parameters used. Different definitions of clinical outcomes yielded differences in the results. It was also observed that the use of different parameters led to various therapeutic outcomes (shown in Table 2). However, despite the fact that the use of different parameters may yield different clinical outcomes, the results still indicate that coordination function, especially in the lower limbs, can reflect the clinical outcome and cognitive or global function for AD (shown in Table 3 and Table 4). However, to the best of our knowledge, there has so far been a lack of consensus around how to define AD clinical outcomes, and our definitions have been used extensively in our previous publications [34,35].

We have reported that coordination function is associated with clinical outcomes, but we did not quantify the findings to report an algorithm that can be used for this assessment. We intend to report on an appropriate algorithm after collecting more samples to ensure its accuracy.

## 5. Conclusions

Evaluation of cognitive and non-cognitive outcomes for AD could be considered simultaneously to have more comprehensive measurements. The use of a kinetic depth sensor to determine the coordination between joints, especially in lower limbs, can significantly reflect the global functional and cognitive outcomes in AD. Such evaluations could be examined by referring to other recognized biomarkers to provide more objective evidence for future AD management.

## Figures and Tables

**Figure 1 brainsci-12-01370-f001:**
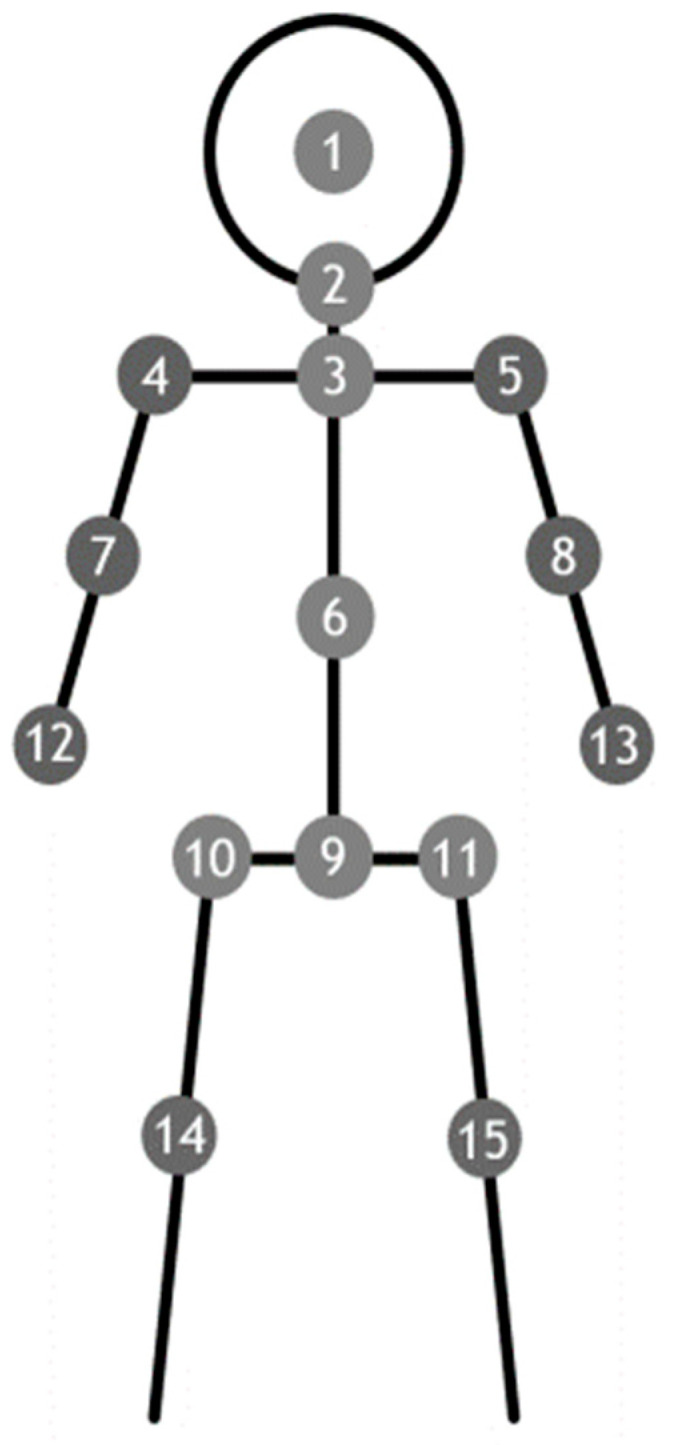
Signals from the 15 marked joints were measured using a kinetic depth sensor. (1) Head, (2) neck, (3) shoulder center, (4) left shoulder, (5) right shoulder, (6) trunk center, (7) left elbow, (8) right elbow, (9) hip center, (10) left hip, (11) right hip, (12) left hand, (13) right hand, (14) left knee, and (15) right knee.

**Table 1 brainsci-12-01370-t001:** Demographic characteristics of patients with Alzheimer’s dementia.

Characteristic	N = 113		
Age (Mean ± SD), Years	78.9 ± 6.9		***p*-Value**
Sex, Female, n (%)	75 (66.4%)		
Education (mean ± SD) years	7.6 ± 5.0		
APOE4 (+)(n/N, %)	37/111 (32.7%)		
	1st year	2nd year	
MMSE (mean ± SD)	17.5 ± 5.0	16.4 ± 5.4	0.001
CASI (mean ± SD)	57.7 ± 16.4	54.1 ± 18.7	<0.001
CDR-SB (mean ± SD)	5.8 ± 2.6	6.6 ± 3.0	<0.001
CDR			<0.001
CDR0.5, n (%)	31(27.4)	21(18.6)	
CDR1.0, n (%)	75(66.4)	77(68.1)	
CDR2.0, n (%)	7 (6.2)	13(11.5)	
CDR 3.0, n (%)	0	2(1.8)	

APOE 4 (+): apolipoprotein epsilon 4 carrier; CDR: clinical dementia rating; CASI: cognitive ability screen instrument; MMSE: mini-mental status examination; CDR-SB: sum of boxes of CDR; SD: standard deviation.

**Table 2 brainsci-12-01370-t002:** Characteristics of clinical outcomes for patients with Alzheimer’s dementia as measured using different parameters.

	Parameter (N = 113)							
	CDR		CASI		MMSE		CDR-SB	
Outcome n (%)	Improved 94 (83.2%)	Worse 19 (16.8%)	Improved 44 (38.9%)	Worse 69 (61.1%)	Improved 49 (43.4%)	Worse 64 (56.6%)	Improved 50 (44.2%)	Worse 63 (55.8%)
Age (mean ± SD), years	79.2 ± 6.6	77.0 ± 8.0	80.0 ± 5.9	78.1 ± 7.4	80.4 ± 6.3	77.7 ± 7.1	80.4 ± 6.3	77.7 ± 7.2
Sex, female n (%)	62 (66.0%)	13 (68.4%)	31(70.5%)	44 (63.8%)	32 (65.3%)	43 (67.2%)	32 (64.0%)	43 (68.3%)
Education (mean ± SD), years	7.5 ± 5.1	8.1 ± 4.6	6.9 ± 5.0	8.1 ± 5.0	6.4 ± 5.0	8.5 ± 4.9	6.8 ± 4.9	8.3 ± 5.0
APOE4 (+) (n/N, %)	32/92 (34.8%)	5/19 (26.3%)	19/43 (44.2%)	18/68 (26.5%)	17/43 (35.4%)	20/63 (31.7%)	15/49 (30.6%)	22/62 (35.5%)

APOE4 (+): apolipoprotein epsilon 4 carrier; CDR: clinical dementia rating; CASI: cognitive ability screen instrument; MMSE: mini-mental status examination; CDR-SB: sum of boxes of CDR; SD: standard deviation.

**Table 3 brainsci-12-01370-t003:** Association between coordination function and various clinical outcome measurements in the upper limbs.

Clinical Outcome Variables	Mean Ratio of Improved Coordination in Improved Clinical Outcome (Mean ± SD)	Mean Ratio of Worse Coordination in Worse Clinical Outcome (Mean ± SD)	Correlation Coefficient for 2 Ratios	*p*-Value
CASI	0.784 ± 0.124	0.495 ± 0.137	0.350	0.131
MMSE	0.616 ± 0.133	0.458 ± 0.154	0.126	0.595
CDR-SB	0.726 ± 0.126	0.668 ± 0.121	0.241	0.306
CDR	0.516 ± 0.188	0.027 ± 0.010	1.000	<0.001

CDR: clinical dementia rating; CASI: cognitive ability screen instrument; MMSE: mini-mental status examination; CDR-SB: sum of boxes of CDR; SD: standard deviation.

**Table 4 brainsci-12-01370-t004:** Association between coordination function and various clinical outcome measurements in the lower limbs.

Clinical Outcome Variable	Mean Ratio of Improved Coordination in Improved Clinical Outcome (Mean ± SD)	Mean Ratio of Worse Coordination in Worse Clinical Outcome (Mean ± SD)	Correlation Coefficient for 2 Ratios	*p*-Value
CASI	0.844 ± 0.182	0.533 ± 0.233	0.798	0.006
MMSE	0.333 ± 0.174	0.037 ± 0.019	1.000	<0.001
CDR-SB	0.266 ± 0.150	0.030 ± 0.017	1.000	<0.001
CDR	0.511 ± 0.297	0.057 ± 0.033	1.000	<0.001

CDR: clinical dementia rating; CASI: cognitive ability screen instrument; MMSE: mini-mental status examination; CDR-SB: sum of boxes of CDR; SD: standard deviation.

## Data Availability

All data presented in this study are available upon reasonable request.

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
