# Peer review of "Association between Cerebral Coordination Functions and Clinical Outcomes of Alzheimer’s Dementia"

_brainsci, 2022, doi:10.3390/brainsci12101370_

Round 1

Reviewer 1 Report

Comments and Suggestions for Authors

The article discusses the association between motor coordination (especially in the lower limbs) and the diagnosis of Alzheimer’s Disease (AD). Accurate and alternate detection methods for AD is very important. The article proposes a simple methodology, but the description of the analysis is a bit confusing with a lack of equations and detailed descriptions. The results presented in this article could be of great significance to the community, but I do have a few suggestions which would strengthen the article.

1.       The statistical analysis described in lines 160-170 is very brief. It may actually be confusing to the readers how some of the parameters are computed. For example, the mean ratio of coordination. It would be helpful to add equations to describe how those parameters are computed.

2.       It is not clear how the correlation between different angles is computed. I believe it is the correlation between the time series (1200 time points). Again, adding equations will make it easier for the readers.

3.       Talking about correlation values, a pairwise correlation between 20 variables (20 joints) will result in a 20x20 matrix with 380 non-diagonal values. Are the 380 correlations referring to that? If so, the correlation matrix is symmetric, and it only has 190 unique values. A discussion about that would also add more clarity to the readers.

4.       The values in Table 1 seem to be incorrect. For example, for the MMSE score, the results for the 1st and 2nd year [mean (std)] are 17.5 (5.0) and 16.4 (5.4) respectively. A paired t-test between the two for N = 113 will not result in a p-value less than 0.001. Same for CASI and CDR-SB. Were some different statistical tests conducted? If so, please describe those in detail. As it is presented, the results in table 1 seem incorrect and misleading.

5.       The very first sentence of the abstract (lines 19-20) doesn’t seem to be grammatically correct. Please rewrite that.

6.       In line 24, CASI is mentioned twice.

7.       The conclusion section seems to be missing. It just has the lines from the template. 

Author Response

We have responded to the comments as attached file.

Reviewer 2 Report

Comments and Suggestions for Authors

In this manuscript, Yang et al. would analyze the role of movement impairment in patients affected by Alzheimer’s disease, correlating them to the cognitive status of each patient. For this purpose, they tested the cerebral coordination functions by a kinetic depth sensor in lower and upper limbs, showing that the impairment in motor coordination in lower limbs correlated with cognitive impairment. So, authors proposed these measurements as potential biomarkers for AD.

Despite the Authors defined better their aim, there is the need of solving some issues to ameliorate the quality of manuscript.

In the INTRODUCTION, the Authors described the possible mechanism of Beta Amyloid in impairment of motor functions consisting in a destruction of cholinergic system in AD. This mechanism will be better explained, considering also that other Neurodegenerative disorders, such as Amyotrophic Lateral Sclerosis, are characterized by Beta Amyloid aggregates into cytoplasm of motor neurons, that lead cell death and so an impairment of motor functions.

The  “Methods” section described the experimental design in an accurate way. However, it is not indicated in “Statistical analyses” if the Authors tested the distribution of the analyzed data.

The Results are clearly reported. However, it is not clear how to validate these measurements as biomarkers for AD, because the impairment in motor coordination may be the result of aging processes. The manuscript lacks the comparison between AD patients and healthy subjects, despite the fact that authors pointed out that similar experiments were carried out in healthy population. Moreover, the contribution of Apo E genotyping is not discussed, and it is not clear what fluid biomarkers are taken in account. The comparison between cerebral coordination functions and fluid biomarkers would be useful to better define their role as biomarker for AD for diagnostic and/or prognostic purposes.

The conclusion of the study is lacking.

Other minor issues are the following:

24. “the CASI” is repeated two times

28-30: “The changes in coordination function in the lower limbs can 28 significantly reflect the cognitive outcome, MMSE (p<0.001) and CASI (p=0.006), CDR (p<0.001) 29 and CDR-SB (p<0.001), but only the CDR (p<0.001) for the upper limbs.” – Although I understood what Authors would say, it is better to remodulate the sentence to better understand the meaning of this result.

90-91: “Mini-Mental State Examination (MMSE) derived from the Cognitive Abilities Screening 90 Instrument (CASI) [20-22], the CASI,…”- It’s unclear  and seems that there are repetitions.

Author Response

we have responded the comments as attached file.

Round 2

Reviewer 2 Report

Comments and Suggestions for Authors

The Authors answered adequately to suggested comments.